# A Validated Proteomic Signature of Basal-like Triple-Negative Breast Cancer Subtypes Obtained from Publicly Available Data

**DOI:** 10.3390/cancers17162601

**Published:** 2025-08-08

**Authors:** Cristina Furlan, Maria Suarez-Diez, Edoardo Saccenti

**Affiliations:** Laboratory of Systems and Synthetic Biology, Wageningen University & Research, Stippeneng 4, 6708 WE Wageningen, The Netherlands; cristina.furlan@wur.nl (C.F.); maria.suarezdiez@wur.nl (M.S.-D.)

**Keywords:** biomarkers, classification, clustering, molecular sub typing, protein–protein interactions

## Abstract

Basal-like breast cancer (BLBC) is an aggressive subtype with poor patient outcome. Using proteomic data from two cohorts, this study identified two distinct BLBC subgroups based on differential protein expression. Key findings include upregulation of spliceosome components, alteration of splicing activity and involvement of collagen proteins.

## 1. Introduction

Breast cancer is the most commonly diagnosed malignant tumor in women, and it is the second most frequently diagnosed cancer and, according to 2022 data, the fourth leading cause of cancer-related deaths, with 2.3 million new cases and 670,000 deaths worldwide [1].

Breast cancer encompasses a diverse group of tumors, and the diversity of cancer cell phenotypes, along with the plasticity of the tumor microenvironment, makes classification challenging, particularly regarding treatment responses and disease progression [2]. At the molecular level, breast cancer is highly heterogeneous, posing challenges for diagnosis, treatment decisions, and outcome prediction. Understanding the mechanisms underlying this heterogeneity is crucial to improving diagnosis, prognosis, and therapy.

The identification of breast cancer molecular subtypes is usually performed using signatures [3,4], such as the Prediction Analysis of Microarray 50 (PAM50), which characterizes the expression of 50 genes [5]. Perou et al. initially defined four molecular subtypes based on transcriptomic profiles of 496 genes [6]: luminal-like, human epidermal growth factor receptor (or HER2)-positive (HER2-positive), basal-like breast cancer (BLBC), and normal-like. Luminal-like cancers predominantly express estrogen and progesterone receptors (ER and PR), and were later subdivided into luminal A and B based on proliferation indices, treatment options, and prognosis [7,8]. Luminal A tumors are hormone receptor-positive with favorable outcomes, while type B is associated with a poorer prognosis. In clinical practice, breast cancers are classified into five subtypes based on histological and molecular characteristics: tumors expressing ER and/or PR are considered hormone receptor-positive; those lacking ER, PR, and HER2 are triple-negative breast cancers (TNBCs). TNBCs tend to have worse outcomes and fewer treatment options, while HER2-positive tumors, though aggressive, can be targeted effectively. When gene expression signatures are unavailable, immunohistochemical [9,10] staining of biopsies can be used to assess ER, PR, and HER2 levels to guide therapy [8].

Molecular classification has enabled personalized therapies [2,11], and survival rates have improved over the past two decades [12]. Still, the TNBC subgroup has the lowest survival rate, ranging from 6 to 12. TNBC is highly heterogeneous, with multiple subgroups identified based on molecular and genetic differences [7]. These include mesenchymal-like and claudin-low TNBC [13,14,15], each with distinct features and clinical outcomes [16].

Basal-like breast cancer (BLBC) is a highly aggressive molecular subtype marked by strong expression of genes found in the basal epithelial layer of the mammary gland [17]. It is characterized by high-grade tumors, elevated mitotic activity, central necrotic or fibrotic areas, and prominent lymphocytic infiltration, and occurs more often in younger women (≤40 years) [18]. Patients with BLBC generally have a poor prognosis, with shorter disease-free and overall survival [19,20].

BLBC constitutes between 12.3% and 36.7% of breast cancer cases (see [17] and references therein). Most BLBCs are TNBCs (BL-TNBC) [6,7], though up to 25% are not (BL-nTNBC) [21] and may express low levels of hormone receptors or HER2. Approximately 50–75% of TNBCs have a basal-like phenotype [22], and 56% to 90% of TNBC cases share gene expression profiles with BLBC [23]. Both BL-TNBC and BL-nTNBC express basal cytokeratins, including CK5/6, CK14, and CK17 [24]. BLBC tumors generally lack ER or HER2 receptors, which renders treatments such as aromatase inhibitors (targeting ER), or trastuzumab (targeting HER2) ineffective [25,26].

Thus, BLBC adds to the challenge of finding effective, subtype-specific treatments. Ignoring TNBC diversity can affect clinical trial interpretations and limit the applicability of results [27].

Due to tumor heterogeneity that extends beyond DNA or RNA profiles, gene expression-based signatures like PAM50 are not always reliable for guiding treatment, particularly in aggressive TNBC cases [28,29]. Moreover, gene expression changes do not always align with protein abundance, which more directly reflects functional biological changes [30]. To address these limitations, classifications based on protein expression profiling have been proposed to better capture the functional phenotypic differences driving breast cancer heterogeneity [4,31,32]. Exploring basal-like TNBC heterogeneity at the protein level can reveal underlying biology, identify therapeutic targets, and support personalized treatment strategies, as proteins are the primary functional molecules in cells.

Proteomic approaches have become increasingly important in classifying functional subtypes and stages of breast cancer, understanding its origin, development, aggressiveness, and predicting recurrence [31,33,34,35]. Proteomic data analysis has identified distinct protein expression patterns linked to malignancy, along with pathway alterations associated with the biological and clinical behaviour of each tumor subtype [36,37,38], and response to neoadjuvant treatment [39]. In this study, we analyse proteomic profiles of basal-like triple-negative breast cancer patients to investigate potential heterogeneity within this subgroup.

We used publicly available data from two studies within the Clinical Proteomic tumor Analysis Consortium (CPTAC) [40]. Anurag et al. identified proteogenomic markers linked to chemotherapy resistance and response in TNBC patients [41], while Krug et al. integrated genomic, transcriptomic, and proteomic data to study breast cancer development and progression [42]; in the Anurag et al. study [41], we identified two basal-like TNBC sub-groups with distinct proteomic profiles, which we validated using the Krug et al. [42] dataset. The distinguishing proteomic signature includes several interacting proteins, some previously unrecognized in cancer, suggesting subgroup-specific splicing dysregulation and cytoskeletal reorganization. These findings point to novel protein-based signatures that may refine TNBC classification, improve prognosis, and inform targeted therapies.

## 2. Materials and Methods

### 2.1. Experimental Data

Two publicly available proteomics cancer datasets, containing protein abundances measured on tumor biopsies from triple-negative breast tumor patients, were used in this study as Discovery and Validation datasets: see Figure 1 in Results for an overview of the study.

*Discovery dataset:* The study by Anurag et al. [41] originally contained 71 samples from women at least 18 years old, diagnosed with clinical stages of II/III ER-negative and HER2-negative invasive breast cancer. We selected the 30 patients/samples that had been classified as basal-like triple-negative breast cancers. In this dataset, protein measurement was performed by TMT-labeling coupled to MS analysis, followed by quantification, normalization, and filtering on quality, resulting in data for 11,062 proteins. Full details on sample preparation and MS protocols are reported in the original publication [41].

Data was retrieved from the Proteomic Data Commons database [43] with the accession identifiers PDC000408 (TNBC biopsies proteome raw files).

*Validation dataset:* The study by Krug et al. [42] included 122 samples from newly diagnosed, untreated patients (stage IIA-IIIC) or undergoing needle biopsy before neoadjuvant therapy. From these, 23 were classified as basal-like triple-negative breast cancers and used for subsequent analysis in the present study. Protein measurement, performed by TMT-labeling coupled to MS analysis, followed by quantification, normalization, and filtering on quality, resulted in the measured abundances of 10,054 proteins. We refer to the original publication [42] for details on sample collection, preparation, and MS experimental protocols. In this dataset, protein abundance data is expressed as two-component TMT normalized $\log_{2}$-ratios of protein abundances in a sample to the common reference sample obtained from 40 tumors, with the ratios normalized by mean centring and standard deviation scaling. The common reference sample consisted of peptide material from all clinical core samples, with an even proportion contributed for each patient. Data can be retrieved from the Proteomic Data Commons database (accession number PDC000120) and from the CPTAC Data Portal [44] (https://proteomics.cancer.gov/data-portal/ (accessed on 31 January 2025)) with accession number S060.

#### Patient and Sample Classification

The classification of molecular sub-types of cancer samples was performed via IHC (immunohisto-chemistry) and the FISH (fluorescence in situ hybridization) assay on cut tissue samples in combination with PAM50 assay. Full details on histochemistry methods can be found in the original publications [41,42].

The samples for this study were selected from the original papers based on being classified as “Basal” by the PAM50 method.

### 2.2. Statistical Methods

#### 2.2.1. Handling of Missing Data and Imputation

Proteins with >25% missing values were removed: 2189 from the Discovery dataset and 881 from the Validation dataset. The Discovery dataset has dimensions of 30×8873 (samples × proteins); the Validation dataset has dimensions of 23×9173. The remaining missing values were imputed using a KNN-based imputation [45] via the knn.impute function from the impute R package [46]. For both the Discovery and Validation datasets containing missing data, K=21 neighbours were used to generate 1000 imputed datasets that were then averaged to obtain the final imputed version of the datasets.

#### 2.2.2. Clustering of Samples

Clusters of patients/samples in the Discovery and Validation datasets were found using *k*-means clustering [47,48] using the Hartigan–Wong algorithm [49]. Different clustering solutions were obtained for k∈[2,3,4,5,6,7,8] with R=1000 different initial random sets.

The optimal number of clusters was determined using two methods: the elbow method [50] and the silhouette method [51]. For the elbow method, the within-cluster sum of squares WCSS was plotted against the number of clusters *k*, and the optimal cluster solution was identified at the value of *k* for which adding more clusters no longer significantly reduced WCSS. The silhouette method evaluates the separation between clusters by assigning each data point a silhouette value ranging from −1 to 1, where values closer to 1 indicate well-defined clusters. A higher average silhouette score across all clusters suggests a better clustering solution. Both approaches indicated that the best clustering solution for both Discovery and Validation datasets is defined for k=2 clusters.

The stability of the cluster solutions was assessed using two different criteria based on sub-sampling, bootstrapping, and data corruption with noise [52]. In the sub-sampling approach, 90% of the samples are randomly selected R=100 times to obtain solutions with k=2 clusters; in the bootstrap approach, R=100 bootstrapped datasets are created (i.e., resampled with replacement, creating datasets with the same size as the original). The sub-sampled and bootstrapped datasets were subjected to *k*-means clustering . The Jaccard similarity index *J* between the clusters obtained from the resampled/bootstrapped and the original cluster was calculated and used to quantify the stability of the clusters as proposed in [53]. Values of J>0.75 indicate stable clusters, and J>0.85 indicate very stable clusters [54]. For the two clusters in the Discovery dataset, we obtained $J_{sub}=0.78$, $J_{boot}=0.73$, for Cluster A and $J_{sub}=0.72$, $J_{boot}=0.70$, for Cluster B. For the two clusters in the Validation datasets, we obtained $J_{sub}=0.83$, $J_{boot}=0.88$ for Cluster A and $J_{sub}=0.83$, $J_{boot}=0.77$ for Cluster B, as implemented in [55].

#### 2.2.3. Matching of Clusters

The clusters found in the Discovery and Validation datasets were matched on the basis of the content of the biological information associated with the centroids (i.e., the 1×P dimensional vector of the means of the relative abundances of the proteins measured in both datasets, P=7639). Operationally, for each cluster, we took the absolute value of the cluster centroids and calculated the upper 75% quantile $q_{75}=0.70,0.36,0.61,0.99$, for clusters 1 and 2 in the Discovery and Validation datasets. Protein enrichment analysis was then performed on the sets of proteins with relative abundance greater than $q_{75}$, for the three Gene Ontology classes, molecular function (MF), cellular components (CCs), and biological processes (BPs) [56,57]. The overlap among the 50 most enriched MF, CC, and BP classes for each cluster centroid was then calculated: clusters were then matched on the basis of the total number of enriched terms taken over all enriched classes.

#### 2.2.4. Random Forest Modeling

Random forest [58] was used to assess the predictive power of the clusters found in the Discovery and Validation datasets. Predictive models were built using sample cluster labels as class label for the classification. Because of class unbalance, undersampling to 95% of the less numerous cluster was used and repeated R=100 times. The model quality measures accuracy, sensitivity, specificity, and the area under the receiver operating characteristic curve (AUROC) were used to assess the quality of the prediction models and were calculated using standard definitions and averaged over 100 repetitions [59]. Here, the cluster labels were used as ground truth for the model performance evaluation.

Statistical significance of the RF model quality was assessed by means of a permutation test. This consists of performing the RF classification on the data after randomly permuting the class labels and repeating the analysis $N_{perm}=1000$ times and collecting the accuracy, sensitivity, specificity, and AUROC values for each of the permuted models. After collecting these values, a Null distribution $Q_{perm}$ containing the $N_{perm}$ values of the permuted quality metrics is created, against which the statistical significance of the original model, expressed as a *p*-value, is calculated as(1)$$\begin{array}{c} p-\mathrm{value}=\frac{1+\#(Q_{0}>Q_{perm})}{N_{perm}}, \end{array}$$
where $Q_{0}$ is the value of each if of the four quality metrics for the original RF classification model, and $\#(Q_{perm}>Q_{0})$ is the number of permuted values in $Q_{perm}$ that are larger than $Q_{0}$. More details can be found in [60]. Random forest modeling was performed using the randomForest R package (version 4.6.14) [61] and in-house developed scripts for permutation testing.

#### 2.2.5. Differential Analysis of Protein Abundances

Protein differential abundance analysis between the two clusters identified in the Discovery and Validation dataset was assessed by means of a two-sided Wilcoxon signed-rank test [62]: a *p*-value <0.05 was considered significant. The two clusters obtained in the Validation data were of very different sizes (15 and 8). To avoid bias due to group unbalance, the largest group was sub-sampled to eight samples, and differential abundance analysis was performed using the sub-sampled largest cluster and the complete smaller cluster: the analysis was repeated R=1000 times. For each protein, the 1000 *p*-values were collected and pooled to a single *p*-value using the harmonic mean approach for combining the *p*-values of dependent tests [63]. This combined *p*-value was used for subsequent analysis. Significant proteins from the Discovery and Validation datasets with $log_{2}FC>0.05$ and $log_{2}FC$ < −0.05 were intersected to select a validated pool of proteins.

#### 2.2.6. Analysis of Sample Metadata

The metadata of tumor samples assigned to different groups using *k*-means clustering (see Section 2.2.2) were compared using a two-sided Wilcoxon signed-rank test [62]. To compensate for the group unbalance of the two clusters in the Validation data, the same resampling and *p*-value combination strategy was used, as in the case of protein abundance analysis described in Section 2.2.5. Correction for multiple testing was implemented using the Benjamini–Hochberg approach [64]. A corrected *p*-value (false discovery rate, FDR) <0.05 was considered statistically significant.

#### 2.2.7. Protein–Protein Interaction Analysis

Protein–protein interactions (PPIs) for validated proteins were derived from the STRING database (version 12.0) [65,66] using the online interface (https://string-db.org/ (accessed on 31 January 2025)). Only physical interactions were considered (i.e., only protein interactions within the same physical complex), with confidence scores larger than 0.9 and a high FDR stringency (FDR < 0.01). Clustering of proteins in the PPI networks was performed using the DBSCAN (Density-based spatial clustering of applications with noise) approach [67], as implemented in STRING with default parameter ϵ=0.3.

#### 2.2.8. Enrichment Analysis

Gene Ontology (GO) enrichment analysis on validated proteins was performed and visualized using clusterProfiler (version 4.12.06) with org.Hs.eg.db (version 3.19.1) and the Benjamini–Hochberg multiple testing correction with the significance set at the corrected *p*-value (FDR) <0.05. gprofiler2 (version 0.2.3) with source organism *Homo sapiens* used to extract the top 50 most enriched GO terms for each cluster in the Discovery and Validation datasets.

### 2.3. Software and Data

All calculations were performed in R (version 4.0.5) [68] and RStudio (version 1.4.1106) [69]. The data and code for analysis is available at https://github.com/esaccenti/TNBCproteomics, accessed on 4 August 2025.

## 3. Results

### 3.1. Basal-like Triple-Negative Breast Cancer Samples of Discovery Dataset Can Be Separated in Two Clear Clusters

We set out to identify a possible protein signature able to further sub-typing basal-like triple-negative breast cancers. We use two individual large proteomics datasets to identify and validate our protein sub-group signature. The overall analysis pipeline deployed in this study is shown in Figure 1.

The clustering analysis of basal-like triple-negative breast cancer samples was performed using the *k*-means method applied to the 30×8873 proteomics Discovery data. Both the elbow method on the WCSS and the silhouette approach indicated the presence of two distinct clusters of basal-like triple-negative breast cancer samples, as shown in Figure 2A.

These clusters were initially validated in terms of stability using both sub-sampling and bootstrapping: the relatively larger values of the average Jaccard index (see Section 2.2.2) indicate clusters that are stable against perturbation of the data, and thus, are potentially relevant.

We then tested whether the separation could be explained simply by other characteristics of the patients than protein levels. Some characteristics of the samples in the two clusters are reported in Table 1. We did not observe any significant enrichment in terms corresponding to the ethnicity of the patients or type of TBNC cancer. Although several characteristics were different between the two groups at the 0.05 level (two-sided Wilcoxon rank sum test), statistical significance disappeared after correction for multiple testing, resulting in FDR>0.05 for all parameters.

### 3.2. Clustering of Basal-like Triple-Negative Breast Cancer Samples in the Validation Dataset Also Yields Two Clusters

With the aim of validating the proteomics signature distinguishing the two sub-groups of TBNC basal patients, we obtained protein abundance data for 23 patients from another study (Validation data [42]) and we applied the same analysis pipeline deployed on the Discovery data (see Figure 1). Clustering analysis applied to the 23×8873 Validation dataset also indicated, in this case, the presence of two clusters (see Figure 2), as suggested by both the elbow and the silhouette methods, whose stability was also positively assessed by means of subsampling and bootstrapping.

For this dataset, a different set of sample parameters (metadata) was available, but, as in the case of the Discovery dataset, we did not observe significant differences between the two groups, as given in Table 2.

### 3.3. Protein Signature Enables Robust Cluster Classification of Basal-like Triple-Negative Breast Cancer Patients

To further assess the relevance of the two clusters, we applied a random forest (RF) classification using cluster labels as a class for predictive modeling. The RF results indicate a very robust classification model for the Discovery protein signature, as shown by the Receiver Operator Characteristic (ROC) curve associated with the predictive models and the model quality metrics presented in Figure 3A,B, first column. The Multi-Dimensional Scaling (MDS) plot obtained from the proximity matrix of the random forest model is shown in Figure 3C and indicates a rather good separation between the samples of the groups. These results corroborate the relevance of the two clusters identified. When applied to the Validation protein signature, the random forest predictive modeling (see Figure 3A) resulted in a weaker predictive model than the one built on the Discovery data, especially for what concerns the specificity, as reported in Figure 3B. However, the MDS plot obtained from the proximity matrix of the random forest model shows good separation between the samples of the two groups, also in the case of the Validation dataset, as shown in Figure 3D.

Two distinct clusters were found in both Discovery and Validation. However, until this point, our analysis was blind to cluster correspondence, and cluster naming (i.e., cluster 1 and 2, see Figure 2) is arbitrary. Thus, it was necessary to find which of the clusters in the Validation dataset corresponded to which cluster was found in the Discovery dataset. This is fundamental to assessing the directionality of protein expression changes between the two groups of patients and highlighting the underlying differences in biology. Clusters were matched based on the size of the overlap of unique enriched Gene Ontology terms within the molecular functions, cellular compartment, and Biological Function classes.

The overlap between the results of the enrichment analyses of the clusters is shown in Figure 4, and indicates that Validation Cluster 1V corresponds, in terms of biological content, to Discovery Cluster 1D, and Validation Cluster 2V corresponds to Discovery Cluster 2V. We will refer to these two clusters as Cluster 1 and Cluster 2 throughout the remainder of the manuscript.

### 3.4. Identification of Reproducible Differential Proteomic Profiles Between Patient Clusters

The differential analysis of the 8873 protein measured on the Discovery dataset (two-sided Wilcoxon rank sum test) revealed 3258 proteins, whose abundance was different between the two groups at the 0.05 level (1529 proteins upregulated and 1729 downregulated in Cluster 1D). These results suggest the presence of a proteomics signature specific to the two clusters, which is not associated with the patient and sample characteristics. Similarly, we performed differential abundance analysis on the set of 9173 proteins available in the Validation dataset. We found 1877 differentially abundant proteins, of which 1360 were upregulated and 517 downregulated in Cluster 1V. We then used the set of differentially abundant proteins found in the Validation dataset to confirm the signature found in the Discovery data and to eliminate possible false positives. Setting a threshold of ±0.5 on the absolute value of the difference between the $\log_{2}$ relative abundance between the two clusters and a threshold of 0.05 on the *p*-value, we identify 256 proteins upregulated in Cluster 1 and 99 proteins downregulated in Cluster 1.

### 3.5. Up- and Downregulated Proteins Associated with Distinct Cellular and Molecular Pathways

Enrichment analysis of the Gene Ontology class (molecular function, cellular compartment, and biological process) was performed on the validated proteomic signature capable of distinguishing patients in Clusters 1 and 2 (see Figure 2) found in both Discovery and Validation data. Enrichment results are given in Figure 5 and Figure 6 for up- and downregulated proteins, respectively. Upregulated proteins show enrichment of cytoskeletal and extracellular matrix organization functions, as well as enrichment for RNA splicing. Downregulated proteins are instead involved in processes related to telomerase RNA localization to nuclear compartments and Cajal bodies, alongside molecular functions involved in protein folding, chaperone activity, and cadherin-mediated binding. Enrichment analysis did not give significant results on terms related to cellular compartment (CC) for downregulated proteins, while it showcases collagen-containing extracellular matrix, cell-substrate junction, spliceosome complex, and actin filament bundle as top five significantly enriched terms for upregulated proteins.

### 3.6. Network Analysis of Differentially Abundant Proteins Reveal Functional Clusters Centered on Collagen and T-Complex Protein 1

Protein–protein interaction networks were built from the sets of the 256 upregulated and 99 downregulated validated proteins, considering only physical interactions, as defined in the STRING database. The interaction network of the upregulated proteins consisted of 256 nodes (proteins) and 26 edges, representing a physical interaction; the average node degree is 0.202, and the average local clustering coefficient is 0.113.

For a random set of proteins of the same size and degree distribution randomly selected from the human genome, the expected number of edges is 10, indicating that the set of upregulated proteins contains more interactions than expected (PPI enrichment *p*-value $8.71\times 10^{-6}$, suggesting that the proteins are biologically related. The network is shown in Figure 7A.

In the network, we mostly observed interactions between two proteins, and a clique consisting of four collagen subunits: COL1A1 (Collagen Type I Alpha 1 Chain, UniProtKB/Swiss-Prot P02452), COL1A2 (Collagen Type I Alpha 2 Chain, P08123), COL3A1 (Collagen Type III Alpha 1 Chain, UniProt P02461), and COL11A1 (Collagen Type XI Alpha 1 Chain, UniProt P12107). Additionally, we identified a hub protein, SNRPG (Small Nuclear Ribonucleoprotein Polypeptide G, UniProt P62308), that interacts with BUD13 (BUD12 homolog, involved in pre-mRNA splicing as component of the activated spliceosome, Q9BRD0), CWC15 (Spliceosome-Associated Protein Homolog, UniProt Q9P013), GEMIN8 (Gem Nuclear Organelle-Associated Protein 8, UniProt Q9NWZ8), SNRPNP70 (Small Nuclear Ribonucleoprotein U1 Subunit 70, UniProt P08621), and ZMAT2 (Zinc Finger Matrin-Type 2, UniProt Q96NC0).

This set of proteins could be aggregated into 12 groups of interacting proteins with well-defined biological functions. Among these, there are two larger clusters, accounting for a U2-type spliceosomal complex and a collagen fibrillar trimer and MET-activated PTK2 (focal adhesion kinase) signaling; see Figure 7B.

The interaction network of the downregulated proteins consisted of 99 nodes (proteins) and 41 edges (interactions); the average node degree is 0.828, the average local clustering coefficient is 0.237, and the number of expected links is 7, with PPI enrichment *p*-value $1.0\times 10^{-16}$. The set of downregulated proteins is thus highly enriched for physical interaction, as shown in Figure 7C.

There is a large clique consisting of seven interaction proteins, the T-Complex 1 TP1 (UniProt P17987), and six chaperonin-containing TCP1 Subunits: CCT2 (UniProt P49368), CCT3 (P49368), CCT4 (P50991), CCT5 (P48643), CCT6A (P40227), and CCT7 (Q99832). This group of interacting proteins is connected with the TUBA1C (Tubulin Alpha 1c, Q9BQE3) e TUBB (Tubulin Beta Class I, P07437) complex.

The clustering analysis, shown in Figure 7D, indicates the presence of seven enriched clusters, the two larger accounting for the positive regulation of the establishment of protein localization to telomere, the folding of actin by CCT/TriC, and chaperonin-containing T-Complex (in PPI Cluster 1) and Aminoacyl-tRNA biosynthesis and Aminoacyl-tRNA synthetase multienzyme complex in Cluster 2, involving the clique consisting of DARS1 (Aspartyl-TRNA Synthetase 1, P14868), and IARS1 (Isoleucyl-TRNA Synthetase 1, P41252) KARS1 (Lysyl-TRNA Synthetase 1, Q15046).

## 4. Discussion

The proteome of triple-negative breast cancer has been previously investigated to discover molecular features specific to the subtype and derive diagnostic and prognostic signatures [74,75,76,77,77]. In this study, we further investigated the proteome of TNBC using two publicly available datasets [41,42], focusing on the basal subtype.

Cluster analysis revealed two distinct subgroups of basal-like triple-negative breast cancers in the Discovery cohort, which were also confirmed in the Validation cohort (Figure 2). Comparing protein expression between subgroups across both cohorts (Figure 1) identified two sets of 255 upregulated and 99 downregulated proteins, some not previously linked to cancer, and enriched for protein–protein interactions (Figure 7C,D).

### 4.1. Upregulated Proteins Contributing to Cluster Separation Are Enriched for Structural and Extracellular Matrix Functions and for RNA Splicing

In the set of upregulated proteins between basal-like triple-negative breast cancer samples, the top enriched GO molecular functions are actin binding, extracellular matrix structural constituent, actin filament binding, and glycosaminoglycan binding (see Figure 5A), with the consistent annotation of GO biological processes like extracellular matrix organization, extracellular structure organization, and external encapsulating structure organization.

Actin and ABPs (actin-binding proteins) are involved in all stages of carcinogenesis, and reorganization of the actin cytoskeleton mediated by ABPs is inherent in invasion and metastasis; actin-binding proteins create a link between the cytomembrane and nucleus, influencing gene expression via the nuclear actin pool [78]. Altered levels of actin-binding proteins have been associated with a poor prognosis in different type of cancers, including breast cancer [78,79].

We observed upregulation of TAGLN3 (UniProtKB/Swiss-Prot: Q9UI15), one of the three isoforms of the TAGLN family (together with TAGLN1 and TAGLN2; for a discussion of TAGLN2’s role in other types of breast cancer, namely ER-negative; see [79]). Because of TAGLNs’ tissue-specific duality in promoting or suppressing tumor growth and cell migration in cancer cells, current research focuses on their possible use as prognostic/diagnostic biomarkers [80].

Extracellular matrix remodeling (ECM) is pivotal in tumor progression and metastasis as tumors exploit ECM remodeling to create a microenvironment that facilitates tumorigenesis and metastasis [81,82]. Characteristics of the extracellular matrix remodeling in breast cancer differ from the ECM of normal breast tissues [83,84]. There are three main groups of ECM proteins: structural proteins such as collagen and elastin, proteoglycans, and glycoproteins [85]. We observed the upregulation of several collagen proteins, mainly of type 1 in the samples in Cluster 1 (see also Figure 7A,B and associated discussion in Section 4.3), which may be associated with tumor invasion and aggressive tumor behavior [82,86,87].

The most significant enriched biological process associated with upregulated proteins is RNA splicing, suggesting dysregulation of the spliceosome complex: we observed upregulation of some key spliceosome proteins and interacting partners like the Small Nuclear Ribonucleoprotein Polypeptide G (SNRPG), which is discussed in Section 4.3. The spliceosome is a complex molecular machine responsible for removing introns from pre-messenger RNA (pre-mRNA) to create a translatable protein, and it plays a crucial role in the regulation of gene expression [88,89]. In most eukaryotes, there are two forms of spliceosomes: the most abundant, the U2-dependent spliceosome, catalyzes the removal of U2-type introns; the less abundant U12-dependent spliceosome splices the rare U12-type class of introns [88,90]. The core spliceosome, along with its regulatory factors, consists of over 300 proteins and five small nuclear RNAs (snRNAs), playing a crucial role in both constitutive and regulated alternative splicing [91]. These snRNAs interact with seven ’Sm’ core proteins and other additional proteins to form small nuclear ribonucleoprotein (snRNP) particles [92]. Though dysregulated RNA splicing is a hallmark of almost all tumor types, our findings highlight a possible difference in the dysregulation level in the basal-like TNBC in comparison to TNBC. A further characterization of the differences in aberrant RNA splicing in the basal-like TNBC could open to additional pharmacological approaches.

### 4.2. Functions
of Downregulated Proteins

The set of downregulated proteins in Cluster 1 is enriched (most significantly) for molecular functions related to ATP (adenosine triphosphate) hydrolysis activity and ATP-dependent protein folding chaperone, protein folding chaperone, and unfolded protein binding (see Figure 6A). ATP hydrolysis is a key process for the maintenance of cell functioning and viability [93], as it involves the catabolic reaction through which energy is released from ATP from the breaking of high-energy phosphoanhydride bonds [93]. This process is more efficient than glycolysis, and in normal cells, energy for metabolic activities is mostly obtained through mitochondrial oxidative phosphorylation (OXPHOS) [94]. In cancer cells, there is a continuous remodulation of the ratios between glycolysis and OXPHOS, of glucose and glutamine, and of glucose/glutamine and fatty acids to yield total ATP [94]. Insufficient OXPHOS, together with elevated glycolysis and operational mitochondrial substrate level phosphorylation, can lead the cell to uncontrolled proliferation, de-differentiation, apoptotic resistance, and ultimately, cancer [95]. Among the downregulated proteins, we observed several mitochondrial proteins (MRPL16, MRPL24, and MRPL37) and TIMM23 (translocase of inner mitochondrial membrane 23 and TOMM40 (translocase of outer mitochondrial membrane 40). Aggressive triple-negative breast cancers are characterized by unique mitochondrial genetic and functional defects [96], and TNBC cells have low mitochondrial respiration in comparison with oestrogen receptor (ER)-positive cells [97].

Enriched molecular functions include the regulation of protein folding, where the phenotype emerges from the genotype through protein folding and protein homeostasis [98]. Protein folding is an ongoing cellular process which is regulated by chaperones [99]. We observed the downregulation of six chaperonin-containing TCP1 Subunits (see also Figure 7 C,D), which are a family of ATP-dependent proteins involved in the folding of unfolded or misfolded proteins [100,101]. The chaperonin-containing TCP-1 (CCT) or TCP1-ring complex (TRiC) is required for the production of native actin and tubulin [102], and thus the downregulation of these proteins, together with the upregulation of ECM proteins, suggests dysregulation and reprogramming of the cytoskeletal network towards cancer progression through the promotion of tumor cell survival, growth, and invasion [103], as the migration and establishment of metastatic colonies requires dynamic cytoskeletal modifications, characterized by the polymerization and depolymerization of actin [104].

Telomerase RNA regulation and localization to Cajal body (see Figure 6B) are the most enriched biological processes characterizing a downregulated group of proteins in Cluster 1. The Cajal bodies are nucleoplasmic structures containing coiled threads of the coilin protein [105]. The interaction of coilin with other proteins enhances several nuclear processes, among which is the modification and assembly of U small nuclear ribonucleoproteins, forming the RNA splicing machinery [105]. The role of the spliceosome in tumoral malignancies has been widely acknowledged [106,107,108,109], as mentioned above: cancer cells undergo significant transcriptome alterations, in part by adopting cancer-specific splicing isoforms. These isoforms and their encoded proteins actively drive cancer progression or contribute substantially to specific cancer hallmarks [110].

Splicing dysregulation has emerged as a novel hallmark of breast cancer, with oncogenic splicing variants of HER2, ER, BRCA1, AIB1, and other tumor- and metabolism-related genes linked to heightened malignancy, poor prognosis, and treatment resistance. Alterations in splicing events have shown promise in predicting prognosis and treatment response in breast cancer patients, highlighting their potential role in precision medicine [111]. For TNBC, prognostic alternative mRNA splicing signatures have been proposed [112].

### 4.3. Dysregulation of Upregulated Interacting Proteins Involves SNRPG, Collagen, and PRC1 Complexes

A central protein in the interaction networks of upregulated proteins, see Figure 7C,D, is the Small Nuclear Ribonucleoprotein Polypeptide G (SNRPG). This is an 8.5 kDa protein, which is involved in pre-mRNA splicing as a core component of the SMN-Sm complex that mediates spliceosomal snRNP assembly, and as a component of the spliceosomal U1, U2, U4, and U5 small nuclear ribonucleoproteins (snRNPs), which are the building blocks of the spliceosome. It is a component of both the pre-catalytic spliceosome B complex and activated spliceosome C complexes. SNRPG is also a component of the minor U12 spliceosome; as part of the U7 snRNP, it is involved in histone 3′-end processing (see UniProt P62308 accession info). Altered levels of SNRPG have been found in breast cancer [113] and other types of cancers, and increased levels of SNRPG have been found to be positively associated with disease initiation, progression, and severity [114], and different expression patterns associated with different types of cancers have been suggested to depend on the protein’s overexpression, mislocalization of unassembled protein, or the mislocalization of misassembled protein [115,116].

Thus, SNRPG may contribute significantly to the initiation and progression of cancers, and its activity is regulated by both specific and non-specific protein–protein interactions [115]. Its network of interactions is known to comprise more than 115 interacting partners and 138 different interactions [117]. We found that four interacting proteins of SNRPG are also upregulated in Cluster 1 of basal breast cancer patients with respect to Cluster 2 (see Figure 7A,B): BUD13, CWC15, and SNRNP70 and ZMAT12, forming a cluster of interacting proteins enriched for the U2-type spliceosomal complex. Once more, this indicates a possible specific dysregulation of spliceosome activity between these two groups of patients.

Several studies have shown that certain snRNPs, like SNRNP200, SNRPD1, SNRPE, SNRPB2, SNRPC, and U5 snRNP, are associated with breast cancer progression, prognosis, and potential therapeutic targets, particularly in triple-negative breast cancer [118,119,120]; SNRPC is frequently upregulated in TNBC and associated with poor prognosis [121].

While SNRPG is an essential component of the gene splicing machinery, there is currently no substantial evidence directly linking it to TNBC. The same is true for the other proteins BUD13, CWC15, SNRNP70, and ZMAT12. This may suggest evidence of a novel signature able to distinguish between different subtypes of basal triple-negative breast cancers.

Collagen subunit proteins COL1A1, COL1A2, COL3A1, and COL11A1 have been related to triple-negative breast cancer: COL1A1 expression is elevated in TNBC tissues and is associated with increased tumor stiffness, promoting cancer progression and metastasis, making it an independent prognostic factor [122]. COL1A2 has been linked to reduced overall and recurrence-free survival in breast cancer [123]. COL3A1, which is an essential component of the extracellular matrix, has been identified in cancer-associated fibroblasts within the TNBC tumor microenvironment, suggesting that it may facilitate the metastasis process through specific signaling pathways [124]. Finally, COL11A1 has been associated with poor survival, chemoresistance, and recurrence in breast cancer, suggesting a potential role in TNBC progression [125,126].

The upregulation of collagen is associated with increased tumor invasiveness: in the tumor microenvironment, cancer-associated fibroblasts lead to excessive collagen synthesis and remodeling, resulting in the stiffening of the extracellular matrix. This is partially mediated by crosslinking enzymes like lysyl oxidase (LOX), which enhance tissue rigidity and promote integrin-mediated signaling. This activates downstream pathways, including focal adhesion kinase, Src, and Rho/ROCK, causing the rearrangement of the cytoskeleton, and the increase in contractility and motility of tumor cells [127]. In aggressive tumors, collagen fibers undergo spatial reorganization, aligning and leading to contact guidance, facilitating cell migration [128].

Among the upregulated proteins interacting directly, three are of particular interest: the YY1 Ying and Yang 1 Transcription Factor (UniProtKB/Swiss-Prot: P25490), RYBP (RING1 and YY1 Binding Protein, Q8N488), and PCGF5 (Polycomb Group Ring Finger 5, Q86SE9). The transcription factor Yin Yang 1 (YY1) is a ubiquitously expressed protein that plays a crucial role in various biological processes, including embryogenesis, differentiation, replication, and cellular proliferation. Depending on its interactions with other transcription factors and co-factors, YY1 can function as both a transcriptional activator and repressor [129]. The role of YY1 in cancer has been widely studied [130], and YY1 overexpression has been reported in malignant tissues and is linked to invasion, metastasis, and poor prognosis across multiple cancer types [130]. However, its precise role and the consequences of its up- and/or downregulation in breast cancer are not clear. In breast cancer, elevated YY1 levels have been found to lead to FEN1 downregulation, increasing cancer cell sensitivity to mitomycin C or Taxol [129]. Depletion of YY1 suppresses clonogenicity, migration, invasion, and tumor formation in breast cancer cells; ectopic YY1 expression in non-tumorigenic epithelial cells can enhance their migration and invasion capabilities [131]. Yet, other studies have found that increased expression of YY1 in breast cancer cells inhibited cell proliferation, foci formation, and tumor growth in nude mice [132], and that YY1 can suppress the growth of various tumor cell types, including breast [133]. RYBP is a key interaction partner of YY1, and it is also a component of the Polycomb repressive complex 1 (PRC1), a well-known chromatin-modifying complex that monoubiquitinates histone H2A, thus repressing gene expression during development and in cancer [134]. In breast cancer, RYBP overexpression has been associated with tumor suppression by inhibiting cell proliferation and metastasis via the regulation of proteins such as cyclin A, cyclin B1, and E-cadherin [135]. RYPB can stabilize the tumor suppressor protein p53 by modulating MDM2, leading to enhanced p53 activity and induction of cell-cycle arrest and apoptosis [136]. As a result of the interaction between YY1 and RYBP, the role of RYBP cannot be clearly explained, and some studies suggest that RYBP may support tumor progression by stabilizing PRC1 and repressing tumor suppressor genes, and its function seems dependent on the cellular context [137].

PCGF5 is also a component of the PRC1 (non-canonical PRC1) complex: overall, PRC1 components can interact with oestrogen receptor alpha (ERα), and the factor FOXA1 in ER-positive breast cancer cells, as well as with BRD4 in triple-negative breast cancer cells [138]. However, PCGF5 is not prognostic in breast cancer according to the Human Protein Atlas [139].

### 4.4. TCP1, Microtubule, and ARS Complexes Are Affected by Downregulated Proteins

Among the downregulated proteins, many belong to the same complex that is in turn affected. The chaperonin-containing TCP1 (CCT) complex, also known as the T-complex protein 1 (TriC), consists of eight subunits (CCT1–CCT8), and assists in the folding of key oncogenic proteins, including actin and tubulin [140]. Several CCT subunits, such as CCT2, CCT3, CCT4, CCT5, CCT6A, and CCT7, have been implicated in cancer progression. CCT proteins are overexpressed in multiple cancers, including breast cancer, where they contribute to cytoskeletal organization, cellular migration, and invasion [141]. Overexpression of CCT2, in particular, has been associated with poor prognosis in TNBC due to its role in promoting oncogenic signaling pathways [142].

Microtubules, composed of α-tubulin and β-tubulin heterodimers, are essential for cell division and intracellular transport. The TUBA1C and TUBB proteins are vital components of the microtubule network and are frequently dysregulated in cancer [143]. TUBA1C has been reported to promote tumorigenesis by modulating microtubule dynamics and influencing mitotic spindle assembly [144]. Similarly, TUBB expression is altered in taxane-resistant TNBC, leading to chemotherapy resistance [145]. Targeting these tubulin proteins has emerged as a potential therapeutic approach for TNBC [146].

Aminoacyl-tRNA synthetases (ARSs) are essential enzymes responsible for charging tRNAs with their respective amino acids. The disruption of ARSs, including DARS1, IARS1, and KARS1, has been associated with cancer progression [147]. Aspartyl-tRNA synthetase (DARS1) plays a crucial role in protein synthesis and cellular metabolism. Studies have indicated that DARS1 is upregulated in several cancers, including TNBC, where it supports tumor growth and survival [148]. Isoleucyl-tRNA synthetase (IARS1) is involved in protein translation and has been linked to cancer cell proliferation. While ARSs may be involved in tumorigenesis [149], there is no evidence linking IARS1 with breast cancer, although it has been included in a signature correlating with prognosis in hepatocellular carcinoma [150].

Lysyl-tRNA synthetase (KARS1) gene codes for protein KRS that is a prognostic marker in head and neck squamous cell carcinoma, lung adenocarcinoma kidney, renal clear cell carcinoma [151], and a novel post-operative monitoring and diagnostic biomarker for CRC [152]. KRS has been shown to participate in oncogenic signaling pathways. It interacts with key proteins involved in tumor progression and metastasis, such as the 67 kDa high-affinity laminin receptor (67LR) [153]. In recent years, it has become an interesting target for drug discovery [154]. The role in TNBC is not yet clear, and thus, our analysis points to a possible implication of KRS in a group of metastasis induction in basal-like triple-negative breast cancers and a possible target.

## 5. Conclusions

This study investigated the proteome of basal-like triple-negative breast cancer, using two publicly available datasets, identifying two distinct subgroups within this aggressive cancer subtype. Analysis revealed 256 upregulated and 99 downregulated proteins significantly enriched for interactions, some of which had not been previously associated with cancer. A key finding was the involvement of the spliceosome in TNBC, particularly the protein SNRPG, which was upregulated, along with four of its interacting partners (BUD13, CWC15, SNRNP70, and ZMAT12), suggesting the potential dysregulation of splicing activity between TNBC subgroups. Additionally, collagen proteins (COL1A1, COL1A2, COL3A1, COL11A1) were linked to tumor progression and metastasis, while chaperonin-containing TCP1 (CCT) complex proteins and microtubule-associated proteins (TUBA1C, TUBB) were implicated in cytoskeletal organization and chemotherapy resistance. Aminoacyl-tRNA synthetases (DARS1, IARS1, KARS1) were also identified as potential contributors to TNBC progression.

These findings highlight novel molecular signatures and potential mechanisms driving basal-like TNBC heterogeneity; however, as in any exploratory study, some limitations must be acknowledged when interpreting these results for their clinical and biological relevance.

The sample size of basal-like TNBC patients, although derived from large cohorts, becomes limited when subdivided into clusters, potentially affecting statistical power, leading only to the detection of larger effects while missing possibly biologically interesting variations. While both clustering and differential abundance analysis were mutually validated, further experiments would be needed to confirm the functional roles of the identified proteins. This study could further benefit from the integration with transcriptomic and/or genomic data that was not available for one of the cohorts used; this could provide a more comprehensive view of the molecular mechanisms. Furthermore, the prognostic and therapeutic relevance of these subgroups could not be validated in the absence of longitudinal and treatment response data.

Given the promising results obtained from the analysis of the protein–protein interaction networks, it could be interesting to explore the patterns of correlation that can be experimentally estimated from the measured protein abundances in the four clusters and compared across different clusters. Although correlations are incomplete proxies of physical and biochemical interactions, they can pinpoint functional relationships, such as coregulation, participation in shared pathways, or membership in protein complexes, helping to map cellular processes and identify key regulatory hubs. When comparing different groups or conditions, changes in correlation patterns (and hence in the topology of the networks that can be inferred from them) can uncover dysregulated networks, characterize tumor subtypes, and guide biomarker discovery or therapeutic targeting.

Overall, the findings presented in this study suggest the existence of a novel molecular signatures that could improve TNBC classification, prognosis, and potential therapeutic targeting.

## Figures and Tables

**Figure 1 cancers-17-02601-f001:**
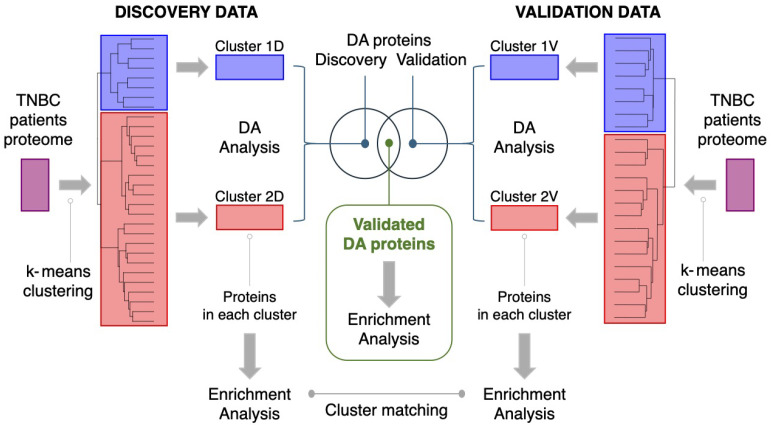
Data analysis pipeline. Clustering was applied to both the Discovery [41] and Validation data [42] (see Section 2.2.2). Two clusters were found in each of the two datasets (clusters 1D and 2D in the Discovery and 1V and 2V in the Validation dataset). Cluster identities were matched on the basis of Enrichment analysis of the 25% most abundant proteins (see Section 2.2.3). Differential analysis (DA) of protein abundances between the two clusters was performed (see Section 2.2.5). A validated set of dysregulated proteins was defined as the intersection of the sets of differentially abundant proteins between the two clusters in the Discovery and Validation data.

**Figure 2 cancers-17-02601-f002:**
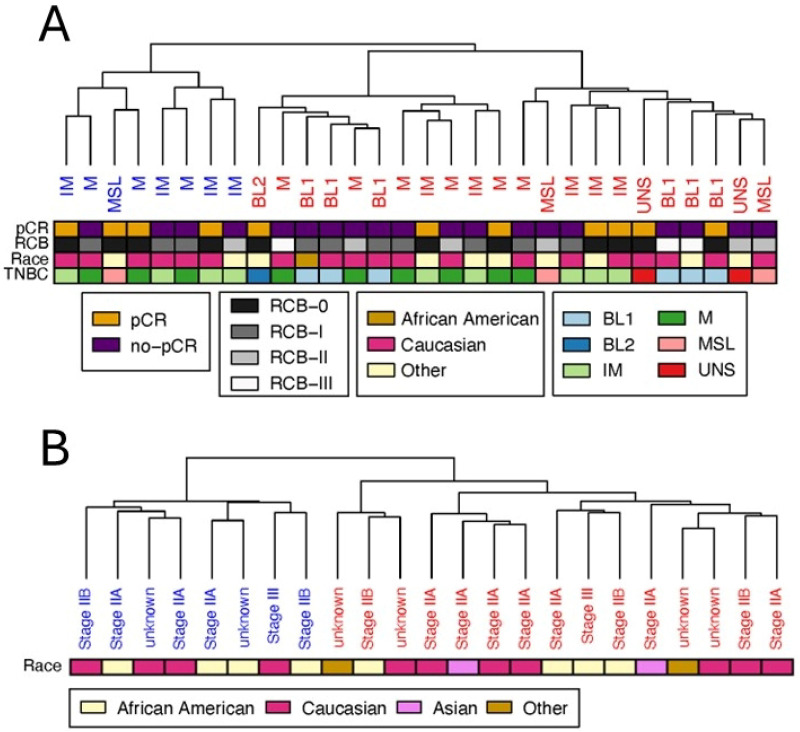
Hierarchical clustering of the basal-like triple-negative breast cancer samples. (**A**) Clustering of samples from the Discovery dataset [41], containing the abundance measurements of 8873 proteins on 30 samples classified as basal-like. Red and blue colors identify samples belonging to two different clusters. Name relates to TNBC type. Heatmaps report metadata related to pathologic complete response (pCR) as a two-level measurement (presence vs. no presence); Residual Cancer Burden (RCB) as a 0-III scale parameter; race of patients expressed as a three-factor variable (African American, Caucasian, or Other), and TNBC type (TNBC) identified by basal-like 1 (BL1), basal-like 2 (BL2), immunomodulatory (IM), mesenchymal (M), mesenchymal stem-like (MSL), and unstable (UNS). (**B**) Clustering of samples from the Validation dataset [42], containing abundance measurements of 9173 proteins on 23 samples. Red and blue colors identify samples belonging to two different clusters. Name relates to tumor staging (IIA-IIB-III-Unknown). Heatmap reports metadata related to race of patients expressed as a four-factor variable (African American, Caucasian, Asian, or Other).

**Figure 3 cancers-17-02601-f003:**
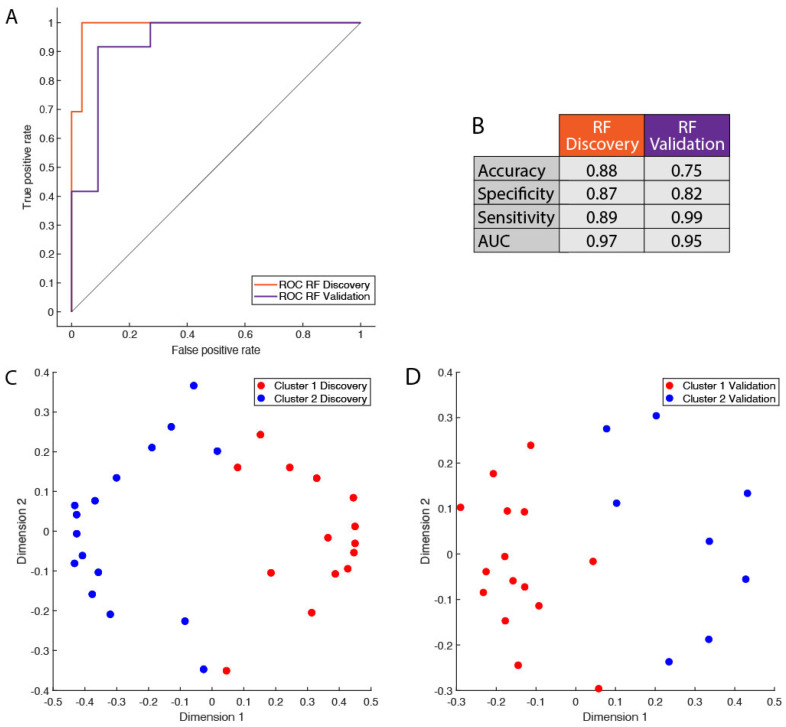
Predictive modeling of basal-like triple-negative breast cancer samples using random forest classification. Sample cluster membership derived from *k*-means clustering was used as class label to build classification models to assess the predictive power of the clustering solution obtained on both Discovery (see Figure 2A) and Validation datasets (see Figure 2B). (**A**) Receiver Operator Characteristic (ROC) curve associated with the classification models in orange for the Discovery dataset and in violet for the Validation dataset; (**B**) summary of the model quality metrics; Multi-Dimensional Scaling (MDS) plots obtained from the proximity matrix of the random forest model for (**C**) Discovery data and (**D**) Validation data, where Cluster 1 is in red and Cluster 2 is in blue.

**Figure 4 cancers-17-02601-f004:**
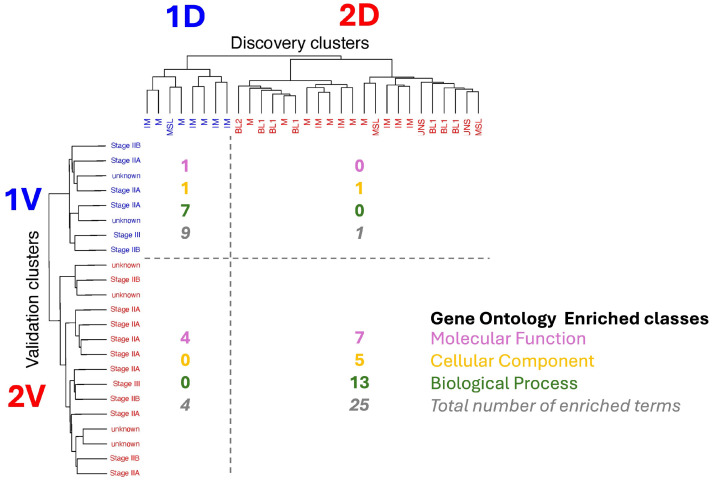
Biology-informed matching of the clusters found in Discovery and Validation datasets (see Figure 2) based on Gene Ontology enrichment. Validation Cluster 1V corresponds, in terms of biological content, to Discovery Cluster 1D (9 unique terms enriched in common), and Validation Cluster 2V corresponds to Discovery Cluster 2D (25 unique terms enriched). Numbers indicate the number of unique enrichment terms for the three GO classes (MF: molecular function; BP: biological process; CC: cellular compartment). MF refers to activities performed by gene products at the molecular level. BPs are larger biological processes carried out through the coordinated action of multiple molecular functions. CCs define the location within a cell where the gene product carries out its function. For more details, see the GO documentation available at geneontology.org.

**Figure 5 cancers-17-02601-f005:**
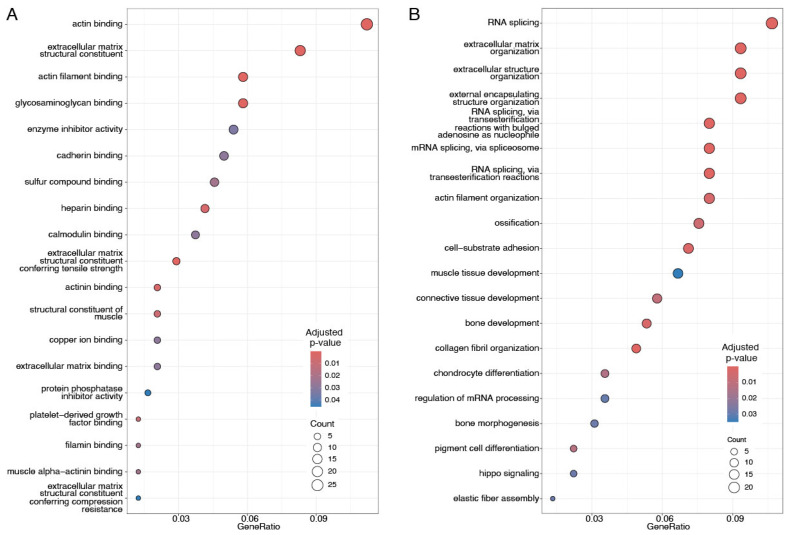
Enrichment analysis of the upregulated proteins between Clusters 1 and 2 for Gene Ontology terms (**A**) molecular function (MF); (**B**) biological process (BP). MF refers to activities performed by gene products at the molecular level. BPs are larger biological processes carried out through the coordinated action of multiple molecular functions. For more details, see GO documentation, available at geneontology.org. Gene ratio refers to the proportion of genes associated with a given GO term, which are also found among the validated 256 upregulated proteins. FDR indicates Benjamin–Hochberg-corrected *p*-values (<0.05). The top 20 most enriched GO categories were selected for visualization.

**Figure 6 cancers-17-02601-f006:**
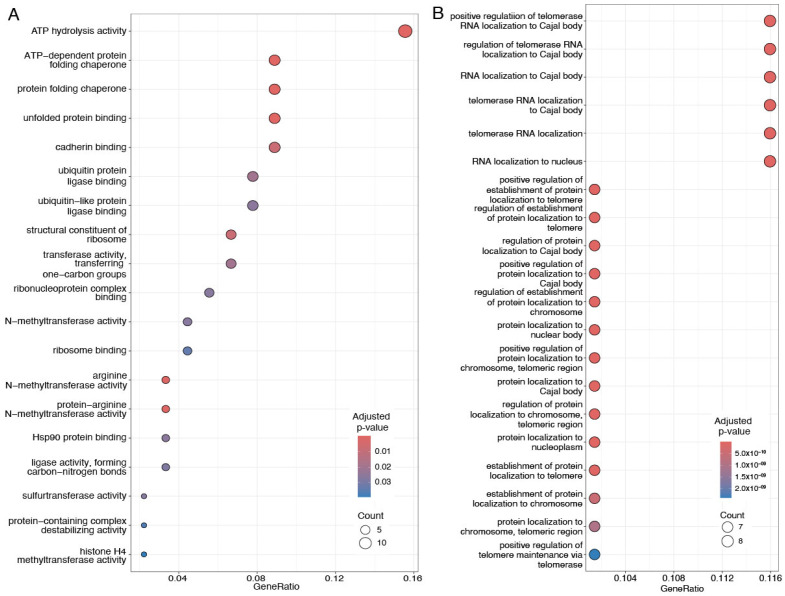
Gene Ontology enrichment analysis of the 99 validated downregulated proteins between Clusters 1 and 2. (**A**) molecular function; (**B**) biological process. For more details, see the caption of Figure 5.

**Figure 7 cancers-17-02601-f007:**
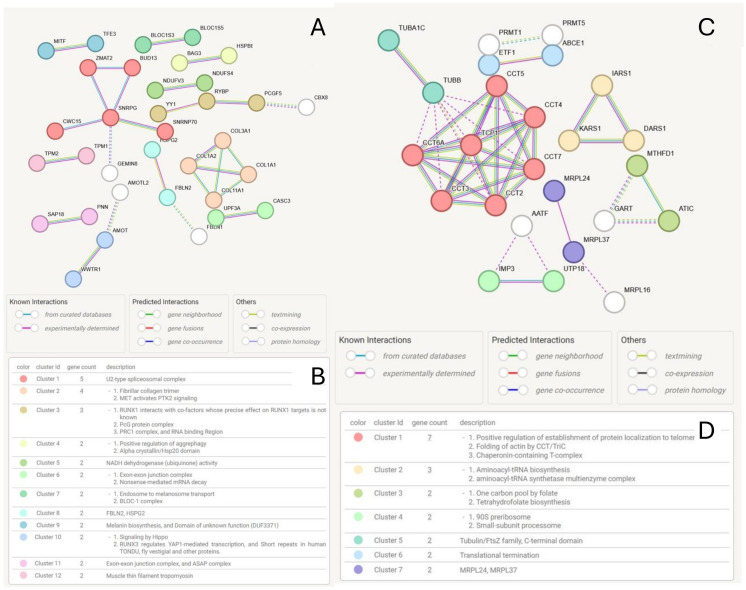
Interaction networks of the up- and downregulated proteins in Cluster 1 of basal triple-negative breast cancer samples obtained from the STRING database. (**A**) Protein–protein interaction network of upregulated proteins. (**B**) Enrichment analysis of the clusters of interacting upregulated proteins. (**C**) Interaction network of downregulated proteins. (**D**) Enrichment analysis of downregulated proteins. Only physical protein–protein interactions (i.e., interactions within the same physical complex) are shown (confidence score >0.9 and FDR < 0.01). Clustering of protein–protein interaction is based on DBSCAN, with the default parameter setting. For more details, see Section 2.2.7. Dashed-lines indicate Ii-cluster edges/interactions.

**Table 1 cancers-17-02601-t001:** Characteristics of basal-like triple-negative breast cancer samples from patients in the two clusters (Cluster 1D and 2D) found in the validation cohort, as shown in Figure 2A. Values are given as medians. Immune profile and microenvironment scores were obtained by ESTIMATE [70], Cibersort [71], and xCell [72]. Protein-based immune modulator scores were calculated as described in [42], taking averages of different types of modulators: immune stimulatory, immune inhibitory, and HLA [73]. For more details, see [41].

Metadata	Cluster 1D	Cluster 2D	Pval	FDR
Average Tumor Content (%) for biopsy	68.125	63.438	0.252	0.378
Chromosomal instability	6.637	4.058	0.003	0.080
Mutation load HG38 v2	668.071	150.438	0.151	0.280
Microsatellite instability score	195.357	24.188	0.013	0.083
Signature 3	0.171	0.152	0.712	0.777
Signature 6	0.016	0.045	0.236	0.378
Signature 15	0.035	0.086	0.324	0.435
Signature 10	0.000	0.006	0.385	0.486
Signature 12	0.044	0.009	0.132	0.280
Signature 4	0.063	0.000	0.011	0.083
Signature 7	0.005	0.004	0.923	0.923
Signature 9	0.009	0.018	0.549	0.628
Signature 13	0.015	0.029	0.326	0.435
Signature 21	0.048	0.005	0.923	0.923
Stimulatory immune modulator proteins	−0.448	−0.092	0.077	0.185
Inhibitory immune modulator proteins	−0.310	0.002	0.146	0.280
HLA immune modulator proteins	−0.821	−0.669	0.506	0.607
ESTIMATE ImmuneScore	1388.4	2200.3	0.038	0.102
ESTIMATE StromalScore	238.5	740.7	0.025	0.083
ESTIMATE TumorPurity	0.654	0.497	0.022	0.083
Cibersort absolute immune score	1.769	2.724	0.017	0.083
xCell ImmuneScore	0.099	0.277	0.019	0.083
xCell StromaScore	0.034	0.046	0.228	0.378
xCell MicroenvironmentScore	0.133	0.323	0.028	0.083

**Table 2 cancers-17-02601-t002:** Characteristics of basal-like triple-negative breast cancer samples from patients in the two clusters (Cluster 1V and 2V) found in the Validation cohort, as shown in Figure 2A. Values are given as medians over 1000 resamplings, as described in Section 2.2.6. Immune profile and microenvironment scores were obtained by ESTIMATE [70], Cibersort [71], and xCell [72]. Protein-based immune modulator scores were calculated as described in [42], taking averages of different types of modulators: immune stimulatory, immune inhibitory, and HLA [73]. For more details, see [42].

Metadata	Cluster 1V	Cluster 2V	*p*-Value	FDR
Chromosome Instability Index	2.765	2.370	0.664	0.800
CIBERSORT AbsoluteScore	1.040	0.889	0.738	0.800
ESTIMATE ImmuneScore	1574.727	1506.019	0.881	0.881
ESTIMATE StromalScore	184.157	−433.674	0.045	0.584
ESTIMATE TumorPurity	0.637	0.709	0.220	0.714
Number of non-synonymous mutations	106.250	115.500	0.399	0.800
Stemness Score	0.704	0.796	0.192	0.714
xCell ImmuneScore	0.094	0.080	0.734	0.800
xCell StromalScore	0.004	0.001	0.300	0.780

## Data Availability

Processed data and code for analysis are available at https://github.com/esaccenti/TNBCproteomics, accessed on 4 August 2025.

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
