# Peer review of "A Validated Proteomic Signature of Basal-like Triple-Negative Breast Cancer Subtypes Obtained from Publicly Available Data"

_cancers, 2025, doi:10.3390/cancers17162601_

Round 1
Reviewer 1 Report
Comments and Suggestions for Authors
This study utilizes publicly available data from two large cohorts of cancer patients to explore the proteomic landscape of BLBC patients. Overall, the study demonstrates good innovation; however, there are several details that need to be revised.
I find the last paragraph of the introduction difficult to understand. This is not a conventional writing style.
The introduction is too long and should be condensed.
The sizes of your tables are inconsistent. Please adjust them according to MDPI’s formatting requirements.
Figure 7: The connections between proteins appear to be limited. The STRING image is not visually appealing—could you consider redrawing it using Cytoscape?
Lines 409–420: I don’t think it is necessary to repeat known experimental results in such detail.
Line 429: Please avoid repeatedly citing the same reference. Additionally, I believe part of this section reflects your own findings, and thus does not require citation.
Line 436: Since your study focuses on triple-negative breast cancer, why are the results related to the prognosis of ER-negative patients?
Line 448: Why is the upregulation of collagen associated with tumor invasion? Please explain this observed phenomenon and the underlying mechanism.
Ensure all figures are referenced properly and provide clear explanations in the captions.
There are too many references. I don't think a 7,000-word manuscript needs so many references. Please delete some of them and keep the number of references to 70 or less.
Many of the references are too old. Please update the references so that they are within the last five years.
The conclusion should not only summarize findings but also highlight the study’s limitations and suggest potential future research avenues.
Author Response
Answers to Comments of Reviewer 1
We thank the reviewer for his/her appreciation and for the time spent on reading and reviewing our manuscript.
Hereafter a point to point rebuttal to his/her comments and suggestions.
- I find the last paragraph of the introduction difficult to understand. This is not a conventional writing style.
We have re-written the last paragraph as:
We used publicly available data from two studies within the Clinical Proteomic Tumor Analysis Consortium (CPTAC) [47 ]. Anurag et al. identified proteogenomic markers linked to chemotherapy resistance and response in TNBC patients [48], while Krug et al. integrated genomic, transcriptomic, and proteomic data to study breast cancer development and progression [49]; in the Anurag et al. cohort, we identified two basal-like TNBC sub-groups with distinct proteomic profiles, which we validated using the Krug et al. dataset. The distinguishing proteomic signature includes several interacting proteins, some previously unrecognized in cancer, suggesting subgroup-specific splicing dysregulation and cytoskeletal reorganization. These findings point to novel protein-based signatures that may refine TNBC classification, improve prognosis, and inform targeted therapies.
- The introduction is too long and should be condensed.
We have shortened the Introduction by ~35%
- The sizes of your tables are inconsistent. Please adjust them according to MDPI’s formatting requirements.
We have uniformed the size of the table font.
- Figure 7: The connections between proteins appear to be limited. The STRING image is not visually appealing—could you consider redrawing it using Cytoscape?
The number of connections reflects what is actually known about the biophysical processes involving the proteins considered. Visual appeal is a rather subjective matter: we personally find the STRING graphical output efficient to convey the information needed.
- Lines 409–420: I don’t think it is necessary to repeat known experimental results in such detail.
Thanks for this suggestion: we have condensed this section to:
Cluster analysis revealed two distinct subgroups of basal-like triple negative breast cancers in the Discovery cohort, which were also confirmed in the Validation cohort (Figure 2). Comparing protein expression between subgroups across both cohorts (Figure 1) identified two sets of 255 up-regulated and 99 down-regulated proteins, some not previously linked to cancer, and enriched for protein–protein interactions (Figure 8A, 8B).
- Line 429: Please avoid repeatedly citing the same reference. Additionally, I believe part of this section reflects your own findings, and thus does not require citation.
We have removed “double” referencing, where appropriate, here and through the text.
- Line 436: Since your study focuses on triple-negative breast cancer, why are the results related to the prognosis of ER-negative patients?
Thanks for this comment: as it is phrased the paragraph concerning TAGLN may have been ambiguous. We have rephrased it as:
We observed upregulation of TAGLN3 (UniProtKB/Swiss-Prot: Q9UI15), one of the three isoforms of the TAGLN family (together with TAGLN1 and TAGLN2; for a discussion of TAGLN2’s role in other types of breast cancer, namely ER-negative, see [85]). Because of TAGLNs’ tissue-specific duality in promoting or suppressing tumor growth and cell migration in cancer cells, current research focuses on their possible use as prognostic/diagnostic biomarkers [86].
- Line 448: Why is the upregulation of collagen associated with tumor invasion? Please explain this observed phenomenon and the underlying mechanism.
The upregulation of collagen is associated with increased tumor invasiveness: In the tumor microenvironment, cancer-associated fibroblasts lead to excessive collagen synthesis and remodeling, resulting in the stiffening of the extracellular matrix. This is partially mediated by crosslinking enzymes like lysyl oxidase (LOX), which enhance tissue rigidity and promote integrin-mediated signaling. This activates downstream pathways, including focal adhesion kinase, Src, and Rho/ROCK, causing the rearrangement of the cytoskeleton, and the increase of contractility and motility of tumor cells [ 134 ]. In aggressive tumors, collagen fibers undergo spatial reorganization, aligning and leading to contact guidance, facilitating cell migration [135].
- Ensure all figures are referenced properly and provide clear explanations in the captions.
We have checked and all Figures and Tables are references and edited the caption for clarity. All captions contain extensive and detailed information. In particular we have expanded the captions of Figures 4, 5, 6, 7 upon merging Figure 7 and 8.
- There are too many references. I don't think a 7,000-word manuscript needs so many references. Please delete some of them and keep the number of references to 70 or less.
We respectfully disagree: we have cited all the work we believe necessary to support the methodological approach used and the discussion of the findings. Before removing references we would need to know which references should be removed and which maintained and the scientific reasoning behind each choice.
Many of the references are too old. Please update the references so that they are within the last five years.
We honestly do not understand the scientific rationale of this request.
11 references of cancer or cancer-biology related studies have been published in the last 5 years and 16 references related to cancer biology are older than 10 years, with the oldest being published in 2001 and referring to the fundamental paper by Sørlie et al., 2001 – Gene expression patterns distinguish tumor subclasses, totalling >14000 references.
We have removed references [1] that being from 1984 was objectively old.
- The conclusion should not only summarize findings but also highlight the study’s limitations and suggest potential future research avenues.
Thanks for the suggestion: we have added a paragraph on (possible) limitations of the study and potential future research avenues.
These findings highlight novel molecular signatures and potential mechanisms driving basal-like TNBC heterogeneity; however, as in any exploratory study, some limitations must be acknowledged when interpreting these results for their clinical and biological relevance.
The sample size of basal-like TNBC patients, although derived from large cohorts, becomes limited when subdivided into clusters, potentially affecting statistical power, leading only to the detection of larger effects while missing possibly biologically interesting variations. While both clustering and differential abundance analysis were mutually validated, further experiments would be needed to confirm the functional roles of the identified proteins. The study could further benefit from the integration with transcriptomic and/or genomic data that was not available for one of the cohorts used; this could provide a more comprehensive view of the molecular mechanisms. Furthermore, the prognostic and therapeutic relevance of these subgroups could not be validated in the absence of longitudinal and treatment response data.
Given the promising results obtained from the analysis of the protein-protein interaction networks, it could be interesting to explore the patterns of correlation that can be experimentally estimated from the measured protein abundances in the four clusters and compared across different clusters. Although correlations are incomplete proxies of physical and biochemical interactions, they can pinpoint functional relationships, such as co-regulation, participation in shared pathways, or membership in protein complexes, helping to map cellular processes and identify key regulatory hubs. When comparing different groups or conditions, changes in correlation patterns (and hence in the topology of the networks that can be inferred from them) can uncover dysregulated networks, characterize tumor subtypes, and guide biomarker discovery or therapeutic targeting.
Overall, the findings presented in this study suggest the existence of novel molecular signatures that could improve TNBC classification, prognosis, and potential therapeutic targeting.
Answers to Comments of Reviewer 2
We thank the reviewer for his/her appreciation and for the time spent on reading and reviewing our manuscript.
We would like to comment on his/her suggestions and on why we decided not to incorporate, at this stage, the suggested analysis:
I suggest (but this is only a non-mandatory suggestion) to generate an 'experimental correlation network' based upon the protein-protein correlation coefficents computed on the relative abundances of protein species observed on the samples.
Firstly, correlations estimated from abundances are (may be) bad proxies for protein-protein physical interaction and bad representatives for the underlying metabolic molecular mechanisms: as such much work has to be done to arrive to a meaningful interpretation of such entities.
Secondly, taking correlation pulling together data from different cluster is, in general, a bad idea, as we showed here:
- Saccenti (2023) What can go wrong when observations are not independently and identically distributed: A cautionary note on calculating correlations on combined data sets from different experiments or conditions. Front. Syst. Biol., 2023 https://doi.org/10.3389/fsysb.2023.1042156
Thirdly, differential correlation and topology analysis is certainly an interesting approach, as suggested by the reviewer, but in this case will entail the (possibly) pairwise comparison of 4 correlation networks and subsequent interpretation and discussion of the findings: this is an amount of work that would require a separate paper itself which we now plan to realise as a follow up for this study
We thank the reviewer for the suggestion, indeed we had not thought of such an approach. We have mentioned this as a possible avenue for new research in the Conclusion section:
Given the promising results obtained from the analysis of the protein-protein interaction networks, it could be interesting to explore the patterns of correlation that can be experimentally estimated from the measured protein abundances in the four clusters and compared across different clusters. Although correlations are incomplete proxies of physical and biochemical interactions, they can pinpoint functional relationships, such as co-regulation, participation in shared pathways, or membership in protein complexes, helping to map cellular processes and identify key regulatory hubs. When comparing different groups or conditions, changes in correlation patterns (and hence in the topology of the networks that can be inferred from them) can uncover dysregulated networks, characterize tumor subtypes, and guide biomarker discovery or therapeutic targeting.
Reviewer 2 Report
Comments and Suggestions for Authors
As first I wish to compliment the authors for a very clear an rigorous data analysis : I particularly appreciated the attention to 'explainability' of the chosen strategy and the conflating of different layers of organization (from microarray based predictor to histochemical descriptors) with the proteome data. The results are in line with the recognition of stroma and cytoskeleton organization on cancer development and the check for possible confounding due to external variables like age are very apt to the point and denote a systemic wisdom of the authors.
While I appreciate the constraining of the protein-protein interaction data coming from STRING to physical interactions so to increase the confidence of the results, I suggest (but this is only a non-mandatory suggestion) to generate an 'experimental correlation network' based upon the protein-protein correlation coefficents computed on the relative abundances of protein species observed on the samples (I suggest the authors to use as threshold the absolute value of correlation putting together up-regulated and dwon-regulated species).
It could be interesting to generate two correlation-based networks (one for each cluster) in order to check for differences and similarities in the wiring pattern of the two networks that will probably give some mechanistic hints. I suggest the authors to focus on 'clustering coefficient' and 'betweeness centrality' of the nodes (proteins) in the two networks (see for example https://www.mdpi.com/1422-0067/23/17/9869) .
Again compliments for a very thorough and rigorous work.
Author Response
Answers to Comments of Reviewer 2
We thank the reviewer for his/her appreciation and for the time spent on reading and reviewing our manuscript.
We would like to comment on his/her suggestions and on why we decided not to incorporate, at this stage, the suggested analysis:
I suggest (but this is only a non-mandatory suggestion) to generate an 'experimental correlation network' based upon the protein-protein correlation coefficents computed on the relative abundances of protein species observed on the samples.
Firstly, correlations estimated from abundances are (may be) bad proxies for protein-protein physical interaction and bad representatives for the underlying metabolic molecular mechanisms: as such much work has to be done to arrive to a meaningful interpretation of such entities.
Secondly, taking correlation pulling together data from different cluster is, in general, a bad idea, as we showed here:
- Saccenti (2023) What can go wrong when observations are not independently and identically distributed: A cautionary note on calculating correlations on combined data sets from different experiments or conditions. Front. Syst. Biol., 2023 https://doi.org/10.3389/fsysb.2023.1042156
Thirdly, differential correlation and topology analysis is certainly an interesting approach, as suggested by the reviewer, but in this case will entail the (possibly) pairwise comparison of 4 correlation networks and subsequent interpretation and discussion of the findings: this is an amount of work that would require a separate paper itself which we now plan to realise as a follow up for this study
We thank the reviewer for the suggestion, indeed we had not thought of such an approach. We have mentioned this as a possible avenue for new research in the Conclusion section:
Given the promising results obtained from the analysis of the protein-protein interaction networks, it could be interesting to explore the patterns of correlation that can be experimentally estimated from the measured protein abundances in the four clusters and compared across different clusters. Although correlations are incomplete proxies of physical and biochemical interactions, they can pinpoint functional relationships, such as co-regulation, participation in shared pathways, or membership in protein complexes, helping to map cellular processes and identify key regulatory hubs. When comparing different groups or conditions, changes in correlation patterns (and hence in the topology of the networks that can be inferred from them) can uncover dysregulated networks, characterize tumor subtypes, and guide biomarker discovery or therapeutic targeting.
Round 2
Reviewer 1 Report
Comments and Suggestions for Authors
My comments were addressed.